# TGFβ1 Secreted by Cancer-Associated Fibroblasts as an Inductor of Resistance to Photodynamic Therapy in Squamous Cell Carcinoma Cells

**DOI:** 10.3390/cancers13225613

**Published:** 2021-11-10

**Authors:** María Gallego-Rentero, María Gutiérrez-Pérez, Montserrat Fernández-Guarino, Marta Mascaraque, Mikel Portillo-Esnaola, Yolanda Gilaberte, Elisa Carrasco, Ángeles Juarranz

**Affiliations:** 1Departamento de Biología, Universidad Autónoma de Madrid, 28049 Madrid, Spain; maria.gallego@uam.es (M.G.-R.); maria.gutierrezperez@estudiante.uam.es (M.G.-P.); marta.mascaraque@uam.es (M.M.); mikel.portillo@uam.es (M.P.-E.); 2Instituto Ramón y Cajal de Investigación Sanitaria, IRYCIS, 28034 Madrid, Spain; montserrat.fernandezg@salud.madrid.org; 3Dermatology Service, Hospital Ramón y Cajal, 28034 Madrid, Spain; 4Servicio de Dermatología, Hospital Miguel Servet, 50009 Zaragoza, Spain; ygilaberte@salud.aragon.es

**Keywords:** cancer-associated fibroblasts, photodynamic therapy, squamous cell carcinoma, TGFβ, resistance, tumor microenvironment, A431, SCC13, fibroblasts, skin

## Abstract

**Simple Summary:**

Photodynamic therapy (PDT) is used for the treatment of in situ cutaneous squamous cell carcinoma (cSCC), the second most common form of skin cancer, as well as for its precancerous form, actinic keratosis. However, relapses after the treatment can occur. Transforming growth factor β1 (TGFβ1) produced by cancer-associated fibroblasts (CAFs) in the tumor microenvironment has been pointed as a key player in the development of cSCC resistance to other therapies, such as chemotherapy. Here, we demonstrate that TGFβ1 produced by CAFs isolated from patients with cSCC can drive resistance to PDT in SCC cells. This finding opens up novel possibilities for strategy optimization in the field of cSCC resistance to PDT and highlights CAF-derived TGFβ1 as a potential target to improve the efficacy of PDT.

**Abstract:**

As an important component of tumor microenvironment, cancer-associated fibroblasts (CAFs) have lately gained prominence owing to their crucial role in the resistance to therapies. Photodynamic therapy (PDT) stands out as a successful therapeutic strategy to treat cutaneous squamous cell carcinoma. In this study, we demonstrate that the transforming growth factor β1 (TGFβ1) cytokine secreted by CAFs isolated from patients with SCC can drive resistance to PDT in epithelial SCC cells. To this end, CAFs obtained from patients with in situ cSCC were firstly characterized based on the expression levels of paramount markers as well as the levels of TGFβ1 secreted to the extracellular environment. On a step forward, two established human cSCC cell lines (A431 and SCC13) were pre-treated with conditioned medium obtained from the selected CAF cultures. The CAF-derived conditioned medium effectively induced resistance to PDT in A431 cells through a reduction in the cell proliferation rate. This resistance effect was recapitulated by treating with recombinant TGFβ1 and abolished by using the SB525334 TGFβ1 receptor inhibitor, providing robust evidence of the role of TGFβ1 secreted by CAFs in the development of resistance to PDT in this cell line. Conversely, higher levels of recombinant TGFβ1 were needed to reduce cell proliferation in SCC13 cells, and no induction of resistance to PDT was observed in this cell line in response to CAF-derived conditioned medium. Interestingly, we probed that the comparatively higher intrinsic resistance to PDT of SCC13 cells was mediated by the elevated levels of TGFβ1 secreted by this cell line. Our results point at this feature as a promising biomarker to predict both the suitability of PDT and the chances to optimize the treatment by targeting CAF-derived TGFβ1 in the road to a more personalized treatment of particular cSCC tumors.

## 1. Introduction

Cutaneous squamous cell carcinoma (cSCC) is the second most common non-melanoma skin cancer (NMSC) after basal cell carcinoma. This malignant form of proliferation of the cutaneous epithelium represents 20 to 50% of all the skin cancers and presents with more mutations compared to other common malignancies. These mutations are mostly caused by the exposure to ultraviolet radiation [1,2,3].

Surgery is still the gold standard for the treatment of cSCC, and chemotherapy with cysplatin, 5-fluorouracil, and taxane is also commonly used [4]. However, cosmetic limitations strongly motivate the use of non-invasive procedures, such as photodynamic therapy (PDT), which has become paramount to treat this malignancy [5]. In addition to the good clinical response to PDT and high cure rate of both invasive and in situ SCC (Bowen’s disease), it is also worth mentioning that PDT is being used as a field treatment for the well-known precursor lesion of cSCC named actinic keratosis, with a successful preventive outcome [5,6,7]. Besides, PDT is clinically approved for the treatment of other forms of NMSC, including basal cell carcinoma and other types of cancer, such as oesophageal cancer [8,9].

PDT is a non-invasive and localized therapy that exerts minimal or no damage and improved healing capacity in the surrounding healthy tissues [10,11]. It consists of a two-step procedure by which the local or systemic administration of a photosensitizer (PS) or its precursor is followed by the irradiation of the lesion with light of a specific wavelength. The interaction between the PS and the light in the presence of molecular oxygen leads to the generation of reactive oxygen species (ROS), which are responsible for the induction of cancer cell death [6,12,13,14]. One of the two most commonly used topical drugs approved for the treatment of NMSC with PDT is 5-methyl aminolevulinate (MAL), the methyl ester form of the aminolevulinic acid (ALA). Both are precursors of protoporphyrin IX (PpIX), an endogenous PS produced as an intermediate metabolite of the heme biosynthetic pathway [15,16,17]. Even though PDT is an excellent treatment for the indicated applications, sometimes the lesions can recur [18].

In the last few years, the perception of cancer biology by the scientific community has changed dramatically, no longer focusing exclusively on tumor cells. In this regard, the complex microenvironment around them, known as the tumor microenvironment (TME), has gained importance [19,20]. The stromal component of the TME is composed of different cell types accompanied by non-cellular components, such as the extracellular matrix (ECM), as well as by other elements, including blood vessels. Overall, this intricate structure makes carcinomas very heterogeneous tissues. Previous studies have shown that the TME plays a key role in tumor initiation, promotion, progression, and metastasis, being directly related to the development of resistances against different therapies, including chemotherapy [19,20,21,22]. Within the TME-resident cell populations, cancer-associated fibroblasts (CAFs) stand out because of their wide effects on both the tumor and its stroma. CAFs are activated fibroblasts associated with the enhancement of the aforementioned stages of carcinogenesis and known to secrete several growth factors and cytokines that promote TME remodelling [19,23]. They have been shown to contribute to tumor progression through the enhancement of cellular migration and the alteration of tumor cell metabolism. Thus, CAFs have been characterized as an effective prognostic factor in different types of tumors [20].

CAFs differ from normal fibroblasts in the expression of different proteins that serve as identity markers [21,24]. Among these, alpha smooth muscle actin (α-SMA), vimentin, CD10, and fibronectin can be highlighted. α-SMA, which has been associated with the enhancement of tumor progression [24,25,26], is a highly conserved member of the actin family with an important role in cell motility, structure and integrity. Vimentin is an intermediate filament protein commonly used as an epithelial-to-mesenchymal transition biomarker, related to cell migration and maintenance of the structure; therefore, its overexpression is associated with an invasive phenotype [27,28]. CD10 is a neutral endopeptidase expressed by both neoplastic and stromal cells and involved in the regulation of cell proliferation and invasion, as well as in the degradation of peptides and cytokines in the TME. Thus, the expression of CD10 is associated with poor clinical outcomes [29,30]. Finally, fibronectin is one of the most abundant ECM proteins which has been highly studied due to its role in the modulation of the ECM. Its overexpression has also been associated with tumor progression [31,32].

Importantly, CAFs secrete different cytokines to the TME, among which transforming growth factor-β (TGFβ) is a major one [32]. This ubiquitously expressed cytokine controls several biological processes through the regulation of proliferation, migration, angiogenesis, apoptosis, or ECM remodeling [33,34]. When TGFβ is secreted and activated in the ECM, it binds to the type II and type I receptors (TβRII, TβRI), triggering the formation of a receptor complex. These steps are mediated by the phosphorylation of both receptor types, and lead to the phosphorylation of the cytoplasmic proteins Smad2 and 3, which then bind to Smad4. The phosphorylated Smad complex subsequently binds specific DNA sequences (Smad Binding Elements, SBEs), regulating the expression of TGFβ target genes [34,35,36]. In addition, the presence of coreceptors in the cell membrane, such as endoglin, can influence the downstream effects of TGFβ [36].

Under pathological conditions, such as cancer, the TGFβ pathway is highly upregulated due to increased levels of the active ligand in the TME [33,34]. Aside from the CAF-derived TGFβ, cancer cells themselves produce this cytokine in variable levels, helping to further recruit and activate CAFs. Altogether, this leads to a global increase in the production and activation of TGFβ [34,35]. In spite of its highly complex role in cancer, it has been described that TGFβ acts as a suppressor in the early stages of tumorigenesis and as a promoter in later phases, revealing the stage-specific effects of this cytokine [33]. More specifically, many studies are focused on TGFβ1 as it is the most abundant isoform [35]. In particular, studies in cSCC have reported an overexpression of TGFβ1 in the non-malignant tissue surrounding the tumor, followed by a decrease in expression after surgical resection [33,35], supporting the previously mentioned tumor-promoting effect. In contrast, a TGFβ1-dependent cell cycle arrest at the G1 phase has been described in both normal epithelium and tumor tissue, potentially supporting a role in tumor suppression [33,34,35]. Interestingly, however, this cell cycle arrest promoted by TGFβ1 has been related to drug resistance in different cancers, including cSCC, as a consequence of the direct repression of specific genes related to cell cycle progression by Smad2/3 proteins [22,37].

In the case of resistance to PDT, CAF-derived TGFβ1 has been identified as an extrinsic factor in different cancers, including cSCC. In the first place, TGFβ1 induces CAFs to produce fibrotic components modifying the composition of the ECM. These desmoplastic matrices can obstruct the PS delivery and therefore impair PDT [32,35]. Secondly, TGFβ1 promotes epithelium-to-mesenchymal transition (EMT) and the acquisition of stem cell-like phenotypes, further contributing to create a fibrotic ECM and to the resistance to therapy. Finally, as previously stated, TGFβ1 produced by CAFs leads to resistance through the cell cycle arrest at G1 phase. Moreover, arrested cells that engage a quiescent state have been identified as drivers of resistance and recurrence in SCC tumors [35,37].

Taking all that into account, the objective of this work is to evaluate the implications of TGFβ1 produced by CAFs in the response of cSCC to PDT, with a particular focus on the molecular mechanisms that can drive resistance to this therapy.

## 2. Materials and Methods

### 2.1. Cell Culture

Two established human cell lines from cSCC, A431 and SCC13, were used [38]. A431 corresponds to a poorly differentiated vulvar tumour, while SCC13 corresponds to a moderately differentiated facial tumour.

Fibroblast primary cultures were obtained from adult patients with in situ SCC (Bowen disease) (T165A and T213A) and invasive SCC (T205A and T223A) in the case of CAFs, and from healthy donors in the case of controls (C1 and C3 were of child origin and C2 was isolated from a healthy adult). Cc, a dermal fibroblast culture from a healthy adult, was purchased from Innoprot (Innovative Technologies in Biological Systems, Derio, Spain).

Human fibroblasts used in the experiments were isolated from 2-mm biopsies from patients with cSCC, with the approval of the Ethics Committee of the Hospital Ramón y Cajal (Madrid, Spain). Once obtained, biopsies were conserved in saline serum for a short time until cultured. For the isolation of fibroblasts, a protocol modified from previously published studies [38,39] was used. Briefly, biopsies were chopped in a Petri dish with 1 mL of dispase II (Sigma, St. Louis, MO, USA) after washing with 1:1 ethanol (Panreac, Barcelona, Spain) and PBS (Phosphate Buffered Saline, Thermo Scientific Inc., HyClone, Rockford, IL, USA). Afterwards, tissues were incubated with 5 mL of dispase II overnight at 4 °C and centrifuged 10 min at 390 g. The pellet was washed with PBS (Thermo Fisher Scientific Inc., Rockford, IL, USA) and centrifuged two additional times, followed by the transference to a flask with DMEM (Dulbecco’s modified Eagle’s medium, high glucose) supplemented with 10% (*v*/*v*) fetal bovine serum (FBS) and 1% antibiotics (penicillin, 100 units/mL; streptomycin 100 mg/mL), all from Thermo Fisher Scientific Inc. (Rockford, IL, USA). When cells had reached 70% confluence, fibroblasts were separated from keratinocytes by differential detachment using 0.05% trypsin (Thermo Fisher Scientific Inc., Rockford, IL, USA).

Cell lines and primary fibroblasts were grown in DMEM supplemented with 10% (*v*/*v*) FBS and 1% antibiotics (penicillin, 100 units/mL; streptomycin 100 mg/mL). Cell cultures were performed under standard conditions of 5% CO_2_, 95% humidity, and at 37 °C. Consecutive passages of cSCC cell lines were achieved by treatment with 1 mM of EDTA/0.25% trypsin (*w*/*v*), while 1 mM of EDTA/0.05% trypsin (*w*/*v*) was used for fibroblasts (both trypsin solutions from Thermo Fisher Scientific Inc., Rockford, IL, USA), followed by centrifugation at 480 g for 5 min at room temperature.

### 2.2. Western Blot

For Western blot (WB), fibroblasts or cSCC cells were seeded in 6-well plates and protein extracts were obtained using RIPA buffer (BioRad, Hercules, CA, USA) mixed with Triton X-100 (pH 7.4, Bioworld, Dublin, OH, USA), phosphatase inhibitors (PhosSTOP EASYpack, Roche, Mannheim, Germany), and protease inhibitors (complete ULTRA tablets Mini EDTA-free EASYpack, Roche, Mannheim, Germany). Afterwards, protein concentration was determined with the BCA Protein Assay Kit (Thermo Scientific Pierce, Rockford, IL, USA). Protein extracts from fibroblast were diluted in Laemmli buffer mixed with β-mercaptoethanol (Bio-Rad, Hercules, CA, USA) and heated for 10 min at 98 °C. Afterwards, extracts were centrifuged for 10 min at 17,968 g at 4 °C. Electrophoresis was performed by using acrylamide/bis-acrylamide gels in denaturing conditions (SDS-PAGE) and transferred to polyvinylidene difluoride (PVDF) membranes (Bio-Rad, Hercules, CA, USA), using a Transblot Turbo system (Bio-Rad, Hercules, CA, USA). Membranes were blocked in skimmed milk in 0.1% TBS-Tween-20, then incubated with primary antibodies against α-SMA (Abcam, Cambridge, UK), vimentin (Abcam), CD10 (Invitrogen, Thermo Fisher Scientific, Waltham, MA, USA), and α-tubulin (Cell Signalling Tecnology, Inc., Danvers, MA, USA) for fibroblasts extracts, as well as HO-1 (Cell Signalling) and vinculin (Sigma-Aldrich, St. Louis, MO, USA) for cSCC extracts. Afterwards, they were further washed and incubated with their corresponding peroxidase-conjugated secondary antibodies (HRP-Goat anti-rabbit IgG and HRP-Goat anti-mouse IgG, Thermo Fisher, Rockford, IL, USA). Protein bands were visualized by chemiluminescence (ECL Plus Kit, Amersham, Little Chalfont, UK) using the high-resolution ChemiDocTR XRS+ system (Bio-Rad) and digitized using Image Lab version 3.0.1 software (3.0.1. version, Bio-Rad Software, Hercules, CA, USA).

### 2.3. Immunofluorescence

For indirect immunofluorescence (IF), fibroblasts were grown on glass coverslips until reaching around 70% confluence and then fixed with formaldehyde (Sigma-Aldrich) diluted to 3.7% in PBS at 4 °C (Thermo Fisher Scientific Inc.). Afterwards, cells were washed three times with PBS and permeabilized with 0.5% Triton X-100 in PBS at room temperature. The cells were then incubated with primary antibodies against α-SMA (Abcam), vimentin (Abcam), CD10 (Invitrogen, Thermo Fisher Scientific Inc.) or fibronectin (Abcam) diluted in 0.5% bovine serum albumin (BSA, Sigma-Aldrich) in PBS for 1 h at 37 °C. After three washes, they were incubated with the secondary antibodies AF546 goat anti-rabbit IgG or AF488 goat anti-mouse IgG (Thermo Fisher Scientific Inc.) for 45 min at 37 °C. Afterwards, cells were washed again and incubated with Höechst 33258 (Sigma-Aldrich) for 5 min at 37 °C for nuclear counterstaining. Finally, the coverslips were mounted with ProLong^®^ (Life Technologies, Carlsbad, CA, USA). The slides were observed using an epifluorescence microscope (Olympus BX61, Olympus Corporation, Shinjuku, Tokyo, Japan) equipped with a HBO 100 W mercury lamp and filter sets for fluorescence microscopy, using blue light irradiation (450–490 nm, BP 490 filter, Olympus Corporation, Shinjuku, Tokyo, Japan) for CD10; ultraviolet irradiation (360–370 nm, UG-1 filter, Olympus Corporation, Shinjuku, Tokyo, Japan) for nuclei; and green light irradiation (570–590 nm, DM 590 filter, Olympus Corporation, Shinjuku, Tokyo, Japan) for α-SMA, vimentin, and fibronectin.

For the quantification of the degree of alignment of fibronectin fibres, IF images were processed using the ImageJ software (version 1.8.0, ImageJ, Bethesda, MD, USA). Briefly, after black to white inversion, the images were convolved and the area of the fluorescent signal was measured. A minimum of four images per condition were included in the analysis.

### 2.4. Measurement of Secreted TGFβ1

To evaluate the production of TGFβ1 by cells in culture, the Quantikine^®^ ELISA Human TGFβ1 Immunoassay (R&D Systems, Minneapolis, MN, USA) kit was used. To obtain the samples, cells were seeded in 24-well plates. When they reached 80% confluence, medium was switched to 250 µL of phenol red-free DMEM (Thermo Fisher Scientific Inc., HyClone, Rockford, IL, USA) supplemented with 1% FBS and 1% antibiotics. After 24 h, supernatants were harvested, centrifuged at 480 g for 5 min and were stored at −20 °C until ulterior analyses. Before performing the quantification assay, latent TGFβ1 present in the supernatants was activated for its detection by incubating the samples with 1 M of hydrochloric acid, followed by neutralization by adding a solution containing 1.2 M of sodium hydroxide and 0.5 M of HEPES. In parallel, cells remaining in the wells were counted using a TC20^TM^ automated cell counter (BioRad) to normalize the TGFβ1 measurements.

### 2.5. Treatment with Conditioned Medium and Exogenous TGFβ1

CM was obtained from fibroblasts seeded and cultured under standard conditions in 75 cm^2^ flasks. When 80% confluence was reached, medium was changed to DMEM with 1% FBS and 1% antibiotics. After 24 h of incubation, the CM was collected for the treatment of tumoral cells. For the treatment with TGFβ1 administered exogenously, lyophilized recombinant human TGFβ1 (Preprotech) was reconstituted in 10 mM of citric acid at a concentration of 1 μg/mL, finally adjusting the pH at 7.4. From this stock, the different concentrations of TGFβ1 required for the treatments were prepared in DMEM with 1% FBS and 1% antibiotics. All TGFβ1 treatments (endogenous and exogenous) were performed during 48 h.

### 2.6. Spheroid Cultures

Spheroid three-dimensional cultures were prepared using A431 and SCC13 cells in a specific medium made of 1:1 DMEM and F12 (F-12 Nutrient mixture, Ham, Gibco, Thermo Scientific Inc., Rockford, IL, USA), 2% B27 serum free supplement (Gibco), 20 ng/mL of EGF (Sigma-Aldrich), 0.4% BSA, and 4 μg/mL of insulin (Gibco). Before seeding the cells, multi-well plates were precoated with 1.2% poly-HEMA (2-hydroxyethyl methacrylate, Sigma-Aldrich) and incubated overnight at room temperature to create a film so that the cells could not adhere to the surface. Then, the cells were plated at a density of 40,000 cells/mL, taking 6 days to form the spheroids.

### 2.7. Photodynamic Therapy and Cell Viability Assessment

For the administration of PDT, 10 mM of stock Methyl-aminolevulinate (MAL) (Sigma-Aldrich) was prepared in deionized sterile water. SCC cells were seeded and grown in culture conditions up to 60–70% confluence. Then, cells were incubated with 0.5 mM MAL for 5 h in DMEM culture medium without FBS and subsequently irradiated with red light doses ranging between 0.6 and 12.2 J/cm^2^. A red-light emitting diode source (WP7143 SURC/E Kingsbright, Angels, CA, USA) with an irradiation intensity of 6.2 mW/cm^2^ and an emission peak at λ = 634 ± 20 nm was used. To minimize light refraction, cells were irradiated from the bottom of the plates. After irradiation, cells were incubated for 24 h in normal conditions before evaluation. This treatment conditions were selected according to previous works performed in our research group as capable to induce cell death in SCC13 and in A431 [10,40].

For PDT resistance studies, A431 and SCC13 cell lines were seeded as described. Then, 24 h after seeding, culture medium was removed, and cells were incubated during 48 h prior to PDT with CM or fresh culture medium containing exogenous TGFβ1, or DMEM with 1% FBS and 1% antibiotics for MAL-PDT. In the case of CM and exogenous TGFβ1 treatments, the concentration of FBS was maintained at 1% throughout the administration of PDT to avoid interferences with the TGFβ1 present in it. Following the pre-treatment, PDT was administered by incubating the cells for 5 h with 0.5 mM of MAL diluted in the corresponding media and subsequently irradiating with the red light doses indicated above. After PDT, the medium was removed and the corresponding media were added to each condition.

To block TGFβ1 signalling, the compounds used were the SB525334 inhibitor of TGFβ1 receptor I (Sigma-Aldrich, kindly provided by Dr. María Luisa Botella, CIB-CSIC, Madrid, Spain) and a TGFβ-neutralizing antibody (R&D Systems). For the administration of SB525334, a 100 mM of stock prepared in DMSO was diluted in DMEM with 1% FBS to reach a final concentration of 1 µM that was added to the cells 15 min prior to the treatment with CM (according to previous works [41]) and it was maintained all along the experiment, performed in the same conditions as indicated above. For the administration of the neutralizing antibody, 1 mg/mL of stock prepared in sterile PBS was diluted in DMEM with 1% FBS. The anti-TGFβ was used at 3 µg/mL, added to the cells 48 h prior to MAL-PDT, and was maintained all along the experiment, as indicated above. Treatment conditions were selected according to previous works [42].

The toxicity of MAL-PDT in cells was assessed 24 h after the phototreatments. The MTT assay was used for bidimensional cultures. To this end, a stock solution of MTT (3-(4,5-dimethylthiazol-2-yl)-2,5-diphenyltetrazolium bromide) (Sigma-Aldrich) in PBS (1 mg/mL) was prepared and diluted in complete medium to obtain a final concentration of 50 μg/mL to be added to the cells and incubated for 3 h at 37 °C. After removing the solution, DMSO was added to dissolve the formazan crystals formed. The absorbance was measured at 542 nm by using the plate reader SpectraFluor, Tecan (Zürich, Switzerland). The results were represented as percentage of cell survival.

In the case of spheroids, they were preincubated with CM for 48 h and subsequently exposed to MAL-PDT. The toxicity of MAL-PDT was determined 24 h after irradiation by propidium iodide-orange acridine assay (PI/OA). The PI red signal served to estimate the % of cells that had lost membrane integrity (dead cells) over the total number of cells forming the spheroid (both alive and dead, all of them stained in green with OA). The fluorescent signal of the compounds was observed using the fluorescence microscope with blue light irradiation (450–490 nm, BP 490 filter) for OA and green light irradiation (570–590 nm, DM 590 filter) for PI. The images obtained were analysed using Image J and represented as an estimation of cell survival rates.

The migration assay was performed after the exposition of spheroids to MAL-PDT in both cases, with or without pre-treatment with CM. Spheroids with their respective medium were transferred to a 12-well plate with coverslips to allow attachment of living cells to the well. Photographs of the development of the spheroids were taken at 0 h, 4 h, and 24 h by using contrast phase microscopy. The cells that survive MAL-PDT are expected to attach to the surface and migrate, while dead cells will remain in suspension.

The diameter of the spheroids was measured using Image J and represented with the GraphPad Prism software (version 6.05, GraphPad Software Inc., San Diego, CA, USA).

### 2.8. Cell Proliferation Assessment

A431 and SCC13 cells were grown on glass coverslips until reaching around 70% confluence and then fixed with 3.7% formaldehide in PBS. Afterwards, cells were washed three times with PBS, permeabilized with 0.5% Triton X-100-PBS at room temperature, and incubated with Höechst for 5 min. Finally, the coverslips were mounted with ProLong^®^. The fluorescent signal of the nuclei was observed under the fluorescence microscope by using ultraviolet irradiation (360–370 nm, UG-1 filter, Olympus Corporation, Shinjuku, Tokyo, Japan). Dividing and non-dividing cells were counted to estimate the mitotic index and the mean +/− SEM was represented.

### 2.9. ROS Production

The production of ROS was evaluated by flow cytometry in both cSCC cell lines. To this end, cells were incubated with 6 µM of dihydro-fluorescein diacetate (DFH-DA) (Abcam) for 1 h, then trypsinized and centrifuged over 5 min at 600 g. The pellet was then fixed with 3.7% formaldehyde for 15 min at room temperature. The fixative solution was then removed after centrifugation at 600 g for 10 min. The pellets were stored at 4 °C until the evaluation by flow cytometry (Cytomics FC500, 2 lasers, Beckman Coulter) (490–530 nm). The fluorescence intensity was determined in 10,000 cells of each population.

### 2.10. Statistical Analyses

All the experiments were repeated at least three times. The statistical analysis and data representation were carried out using the GraphPad Prism software (version 6.05, GraphPad Software Inc., San Diego, CA, USA). A one-way ANOVA test was performed to compare the results obtained at different treatment conditions in comparison with one control condition. A *t*-test was used for the comparisons between two groups. Statistical significance was set at *p* < 0.05.

## 3. Results

### 3.1. Characterization of Cancer-Associated Fibroblasts

#### 3.1.1. Secretion of Endogenous TGFβ1

The first step was to compare the levels of endogenous TGFβ1 produced and secreted by control fibroblasts and CAFs. To this end, the culture medium containing 1% FBS was collected after 24 h of incubation with 80% confluent cell cultures and the levels of TGFβ1 were measured using a specific ELISA kit. The results obtained indicated that most of the CAFs secreted significantly higher levels of endogenous TGFβ1 than controls. Among the controls, C3 (child origin) produced significantly higher levels of this cytokine compared to C1 (child origin), C2 (adult origin), and Cc (commercially available human adult control fibroblasts). Based on these results, we decided to exclude C3 from subsequent analyses and selected C1 as the most representative primary fibroblast control culture. The maximum levels of endogenous TGFβ1 were secreted by CAFs, in particular by T165A, closely followed by T205A and T223A, all of them secreting significantly higher levels than C1. Consequently, the CAF culture T213A was also excluded from subsequent analyses as it secreted low levels of TGFβ1 (similar to those of C1, C2, and Cc) (Figure 1).

#### 3.1.2. CAF Markers

The selected markers were studied by WB and IF. For the preparation of protein extracts, fibroblasts were grown up to 90% confluence and for IF up to 70% confluence. All the results were relativized to the data obtained in fibroblasts isolated from healthy volunteers (in particular, C1).

The first protein studied was α-SMA, a well-known marker whose differential expression is characteristic of CAFs [24]. The data obtained by WB showed that C1 and C2 expressed the highest levels of α-SMA, while C3 expressed much lower levels. Among CAFs, T223A was the only one expressing levels of α-SMA similar to those of C1. On the other hand, the expression of α-SMA in all the other CAFs was significantly lower than in C1, with T213A and T205A showing the lowest α-SMA expression levels (Figure 2a). Similar results were seen by IF, confirming the data obtained by WB (Figure 2b and Appendix A).

The second marker evaluated was vimentin. It is well known for its variable expression levels in CAFs, though its overexpression is generally associated to poorer prognosis [27]. As expected, the results obtained by WB revealed higher expression of vimentin in CAFs than in controls. Within each group, similar levels were found in C1 and C2, while the minimum expression levels of vimentin were found in C3, which showed significantly lower levels than C1. Among CAFs, the maximum expression levels of vimentin were observed in T165A and T223A, followed by T205A and T213A (Figure 2a). These results were in agreement with the data obtained by IF (Figure 2b and Appendix A).

The overexpression of the CD10 endopeptidase is also associated to poor clinical outcomes in human epithelial tumors [29]. The expression of this marker was characterized by WB, revealing that the minimum expression levels of CD10 were observed in C1. The expression level of CD10 was higher in CAFs than in controls, reaching the maximum values in T223A, followed by T165A and T205A. CD10 expression in these three CAF cultures was significantly higher than in C1. The lowest levels of CD10 expression among the CAFs were found in T213A (similar to those of C1) (Figure 2a), in agreement with the low levels of TGFβ1 secreted by this culture, also similar to those of C1. The IF assays confirmed the results obtained by WB regarding the expression of CD10 (Figure 2b and Appendix A).

The last marker studied by IF was fibronectin, as changes in its distribution are associated with ECM remodelling. Fibronectin is known to typically form a mesh in the control cells, and more stretched, parallel fibers in CAFs [26]. The overall distribution of fibronectin observed in the controls was similar to a spiderweb, whereas CAFs showed their fibronectin fibers more aligned and distributed in parallel, with T165A as the most representative example. The intensity per area of fibronectin networks was measured in convoluted images, showing more intensity per area in CAFs when compared to C1 (Figure 2c,d and Appendix A).

Our aim was to study the effect of CM from CAFs on cSCC cells. Based on the results presented above, we selected one control fibroblast culture (C1) and two CAF cultures (T165A and T205A) as the most representative ones to perform the rest of the experiments. We selected C1 as the most representative control fibroblast culture based on its low levels of secretion of TGFβ1, and in agreement with the low levels of expression of CD10 and vimentin together with the expression pattern of fibronectin, in spite of its high levels of expression of α-SMA. Although C2 secreted similar levels of TGFβ1 than C1, the expression of the other markers was less consistent. Among the CAFs, we selected T165A from a patient with Bowen disease and T205A from a patient with invasive SCC, based on their high levels of secretion of TGFβ1 and the consistent pattern of expression of the rest of the markers.

### 3.2. Effect of Conditioned Medium from CAFs on the Resistance of cSCC Cells to PDT

On a step forward, we investigated the effect of CM from CAFs on the resistance of SCC cells to PDT. With this aim, we evaluated the response to PDT of cSCC cells in the presence or in the absence of CM form either control fibroblasts or CAFs. To obtain CM from the selected control fibroblasts and CAFs, the cells were seeded and grown until they reached 80% confluence, followed by the incubation with complete DMEM containing 1% FBS for 24 h before collection. For the treatment of cSCC cells with CM, both A431 and SCC13 cell lines were seeded and cultured following standard procedures, and the cells were incubated with CM from CAFs or control fibroblasts for 48 h prior to PDT. The cells were subsequently exposed to MAL-PDT (0.5 mM of MAL and red light doses ranging from 0.6 to 9.1 J/cm^2^) and the treatment with CM was maintained for another 24 h before performing the appropriate analyses. The corresponding control treatments were carried out in all cases: untreated cells (cells treated with neither MAL nor red light irradiation) as well as cells treated only with MAL (0.5 mM, 5 h) or only with red light (12.2 J/cm^2^). No cell toxicity was detected in any of the control conditions (data not shown).

The response to PDT was assessed in terms of cell survival using the MTT assay. The data obtained revealed that CAF-derived CM induced resistance to PDT in A431 cells (Figure 3a). In this regard, the cell survival rates in response to MAL-PDT observed in the presence of CM from T165A and T205A was significantly higher than in the absence of CM. This revealed a reduction in PDT effectiveness and, therefore, the induction of resistance by components of the CM. This effect was supported by the morphological differences observed by phase contrast microscopy (Figure 3b and Appendix A), with fewer cell death figures observed in A431 cells incubated with T165A-derived CM and exposed to PDT at a dose of 0.5 mM of MAL and 9.1 J/cm^2^, in comparison with A431 cells treated with PDT alone. On the other hand, the CM from C1 did not induce significant differences in the cell viability of cSCC cells in response to PDT compared to in the absence of CM. Interestingly, the incubation of A431 cells with CM from CAFs directly induced a striking change in morphology, with an increased abundance of flattened cells under these conditions even in the absence of PDT (Figure 3b, 0 J/cm^2^).

Conversely, no induction of resistance to PDT was observed in SCC13 cells in the presence of CAF-derived CM. The cell survival rates of SCC13 cells in response to PDT in the presence or in the absence of CM from either control fibroblasts or CAFs were similar in all cases (Figure 3c). The analysis by phase contrast microscopy revealed that the CAF-derived CM alone was also able to directly induce morphological changes in SCC13 cells (Figure 3d, 0 J/cm^2^), albeit a similar frequency of dead cells was observed in response to PDT in the presence or in the absence of CM from CAFs (Figure 3d and Appendix A). Overall, these results indicated that the CM from CAFs did not induce resistance to PDT in this cell line. It is important to highlight that SCC13 cells displayed higher intrinsic resistance to PDT than A431 cells at all doses of PDT (Figure 3a,c, black bars).

In order to confirm that TGFβ1 was the component of the CAF-derived CM that was actively inducing resistance to PDT in A431 cells, the same experiments were carried out in the presence of the SB525334 TGFβ1 receptor inhibitor. For this, A431 cells were treated with 1 µM of SB525334 15 min before the addition of CAF-derived CM. The inhibitor was maintained all along the experiment and the assessment of cell viability by MTT was performed 24 h after MAL-PDT. The data obtained demonstrated that the TGFβ receptor inhibitor effectively prevented the CAF-derived CM from increasing cell viability of PDT-treated cells (Figure 4). Overall, these results provide robust evidence of the role of the TGFβ1 cytokine present in the CM from CAFs in the induction of resistance to PDT in this cSCC cell line.

To further demonstrate that TGFβ1 present in CAF-derived CM was implicated in the resistance to PDT observed in A431 cells, both A431 and SCC13 cell lines were treated during 48 h with increasing concentrations (0.01 ng/mL to 1 ng/mL) of human recombinant TGFβ1. Subsequently, the cells were treated with a fixed dose of PDT that was known to induce around 50% of cell death (LD50). The response to PDT was assessed using the MTT assay. The results showed increased survival of A431 cells in response to PDT in the presence of 0.1 and 1 ng/mL of exogenous TGFβ1 (Appendix A). This indicates that TGFβ1 drives a reduction in PDT toxicity, leading to resistance to the treatment. On the other hand, no resistance to PDT was observed in SCC13 cells, in which PDT induced a similar percentage of cell death in all the conditions evaluated. Altogether, these data recapitulate the results obtained using CAF-derived CM and support that the TGFβ1 produced by CAFs is responsible for the resistance to PDT observed in A431 cells.

Three-dimensional cultures constitute an in vitro tool that more faithfully reproduces the physiological conditions operating in tumors in vivo. Thus, on a step forward, we decided to investigate whether the response to PDT of cSCC cells grown in spheroids was affected by CAF-derived CM. To this end, spheroids were made from each tumor cell line (A431 and SCC13) and grown for 6 days. As for monolayer cultures, MAL-PDT was applied in the presence or absence of CM from either control fibroblasts (C1) or CAFs (T165A and T205A). The MAL-PDT conditions used to treat the spheroids were selected based on previous trials performed in monolayer cultures, taking into account that spheroids are likely to be more resistant to MAL-PDT than monolayer cultures. In this regard, the LD50 in 2D in the absence of CM was set as the minimum red light dose used to treat 3D cultures, and this dose was set as the maximum dose twice. After the exposure of spheroids to MAL-PDT at a dose of 0.5 mM of MAL and red light doses ranging from 6.1 J/cm^2^ to 12.2 J/cm^2^, PI/OA staining was performed in order to estimate the cell viability rate using fluorescence microscopy. The results observed recapitulated the data obtained in the cultures in monolayer. In A431 spheroids, the cell viability after PDT was significantly higher in the presence of CAF-derived CM, revealing a similar resistance effect than that observed in bidimensional cultures. In agreement with this, no differences in the cell viability rate were seen in A431 spheroids treated with PDT in the presence of CM from C1 compared to those treated with PDT alone (Figure 5a and Appendix A).

Unlike A431 spheroids, SCC13 spheroids treated with CM from CAFs or from C1 did not show significant differences in cell viability when compared to those treated with PDT alone (Figure 5b and Appendix A). Overall, the results obtained using spheroids are in accordance with the data obtained using bidimensional cultures and serve to validate these data using a more translational approach.

To further characterize the effects of MAL-PDT combined with CAF-derived CM in A431 three-dimensional cultures, additional parameters were assessed, such as the size of the spheroids and their ability to migrate forming explants. The diameter of the spheroids was measured 24 h after MAL-PDT in each condition, revealing that the spheroids subjected to the treatment with PDT in combination with CAF-derived CM were larger than those treated with PDT in combination with CM from C1 or with PDT alone (Figure 6a). On the contrary, and in agreement with the data of cell viability, the diameter of SCC13 spheroids did not significantly change (Figure 6a). Furthermore, a migration assay was carried out 24 h after treatment. To this end, the spheroids treated with CM from C1 and PDT, CM from T165A and PDT, or with PDT alone were transferred to coverslips. A follow-up of the explant evolution was performed by sequentially taking phase contrast images at different timepoints (0 h, 4 h and 24 h). At 4 h, most of the living spheroids slightly decreased in size and started to adhere to the coverslip. After 24 h from MAL-PDT, living spheroids appeared completely adhered, forming explants, while dead spheroids remained in suspension. The size of the explants indicated an increased migration capacity upon the treatment with CAF-derived CM in combination with PDT, compared to the CM of C1 combined with PDT or with PDT alone. The same results were obtained for both doses of PDT that were tested (Figure 6b). Moreover, no differences were observed between explants treated with C1-derived CM in combination with PDT or with PDT alone. These results support that CAF-derived CM is able to induce resistance to PDT in A431 cells, increasing both cell survival and the migration capacity.

### 3.3. Effect of TGFβ1 on the Proliferation and Cell Cycle Progression of cSCC Cell Lines

In order to understand the mechanisms by which TGFβ1 was differentially inducing resistance in A431 cells, we hypothesized that this cytokine may be driving this effect by exerting a cytostatic activity, which has already been described by other authors. We first investigated the effect of CM from CAFs on cSCC cell proliferation. For these experiments, we primarily used the CM from T165A cells, as the levels of endogenous TGFβ1 produced by this CAF culture were maximal. The results revealed a significant decrease in proliferation in A431 cells in response to the treatment with CM from CAFs (Figure 7a,b and Appendix A), although such a decrease was not observed in the SCC13 cell line. To further characterize this response, we estimated the mitotic index in cell cultures subjected to the treatment with recombinant TGFβ1 compared to the corresponding controls. The results showed an evident decrease in the percentage of dividing cells in both lines in response to exogenous TGFβ1. The decrease in cell division rates was statistically significant at doses of 0.1 and 1 ng/mL of TGFβ1 in A431 and at 1 ng/mL in SCC13 (Figure 7a,c). Altogether, these data suggest that the level of TGFβ1 secreted by CAFs may not be sufficient to affect SCC13 cell proliferation. These results are in agreement with the lack of resistance to PDT observed in SCC13 cells in response to the treatment with CAF-derived CM in combination with PDT.

To better understand how TGFβ1 affects the cell cycle progression and proliferation, we evaluated the cell morphology upon treatment with exogenous TGFβ1. Images of the cell cultures were taken after 48 h of incubation with different doses of the cytokine. The results revealed that the number of rounded and detached cells found in suspension—indicative of cell death [43]—was not increased by the treatment with TGFβ1 (Appendix A). This supports the idea that exogenous TGFβ1 did not induce cell death in any of the cSCC cell lines used in this study. However, a patent morphological change was observed in both cell lines in response to the treatment with TGFβ1, leading to a more flattened cell morphology with increased cytoplasmic surface (Appendix A). This could be associated with an arrest of the cell cycle and therefore a decrease in proliferation.

Additionally, MTT assays revealed significantly reduced levels of absorbance in both cell lines after the treatment with TGFβ1 (Appendix A). As we had ruled out the toxicity of TGFβ1 treatments and in agreement with the morphological changes observed, these results point out to a decrease in cell proliferation induced by TGFβ1.

Taken together, these results support the idea of a potential induction of quiescence by TGFβ1 in cSCC cells.

### 3.4. Molecular Characterization of cSCC Cell Lines

Finally, we wanted to deepen in the characterization of the molecular mechanisms behind the differential responses observed in the two cSCC cell lines used in the study to PDT combined with TGFβ1. First, the endogenous levels of TGFβ1 were measured by ELISA. The results indicated that SCC13 cells produce significantly higher levels of endogenous TGFβ1 compared to A431 cells (Figure 8a). Moreover, we analyzed the expression of key markers that have been related to TGFβ1 and PDT response in other studies. On that account, several authors have reported that TGFβ1 regulates the expression of components of the heme synthesis pathway, such as HO-1. Indeed, significantly higher basal levels of HO-1 expression were found in untreated SCC13 ccompared to A431 cells (Figure 8b, 0 ng/mL TGFβ1). In addition, the treatment with exogenous TGFβ1 induced an increase in HO-1 expression in the A431 cell line at both concentratons tested (0.1 and 1ng/mL), whereas only 1 ng/mL of TGFβ1 was sufficient to significantly increase HO-1 expression in SCC13 cells. The basal ROS levels of both cSCC cell lines were also quantified using DHF-DA, revealing significantly lower basal ROS levels in SCC13 than in A431 cells (Figure 8c). This is consistent with the basal levels of expression of HO-1.

Finally, we demonstrated that the cell viability of SCC13 cells in response to PDT dramatically decreased upon treatment with a TGFβ1-blocking antibody (Figure 8d), supporting the idea that lower basal sensitivity to PDT of SCC13 compared to A431 cells is due to the elevated levels of TGFβ1 secreted by the former. The TGFβ1-neutralizing antibody was added to the culture 48 h before to MAL-PDT and maintained all along the experiment.

## 4. Discussion

The contribution of CAFs to tumor progression and aggressiveness through the secretion of a variety of factors to the TME is well known [19]. Among all these factors, TGFβ has been demonstrated to induce resistance to different therapies [19,20,21]. In this sense, CAFs are known to be a major source of this growth factor [23,31]. More specifically, studies carried out in mouse models with cSCC have proven that TGFβ1 mediates tumor resistance to different chemotherapeutic agents [22,35]. However, no studies have yet addressed this question in the field of PDT, which is becoming a very common therapeutic strategy for the treatment of cSCC given its aforementioned advantages [5,6]. Taking all this into account, our aim was to uncover the implications of CAF-derived TGFβ1 in the modulation of the response of cSCC to PDT and in the development of resistance mechanisms.

As other authors have reported, the most accurate method to identify and distinguish CAFs from normal fibroblasts is the characteristic pattern of expression of specific markers related to their activity and effects in tumor resistance [24,25,26]. On that account, and based on the implications of TGFβ1 in cSCC resistance to other therapies [22], we prioritized the analysis of the endogenous levels of secretion of this cytokine for the selection of T165A and T205A as the most representative CAF cultures, and C1 as a control. On the other hand, despite α-SMA being one of the best known CAF markers, our results were controversial, generally showing lower expression levels of this marker in the majority of CAFs isolated from patients. However, some studies have revealed that the expression of α-SMA may vary among different CAF populations [24], pointing out that its expression levels can show a wide variation in activated fibroblasts that still behave as CAFs. Related to the other markers, the results obtained along the characterization were consistent, since the overexpression of vimentin and CD10 and the parallel alignment of fibronectin fibers are all well-known features of CAFs [26,27,30].

The involvement of CAFs in the resistance to different chemotherapeutic agents has been demonstrated by other authors in prostate, pancreatic, breast, and gastric cancers, among others [41]. Although previous work has also been carried out in SCC [22], our study addresses this question for the first time in regard to PDT. Our findings indicate that CAFs can mediate resistance to PDT in cSCC. Furthermore, this resistance was necessarily induced by secreted factors, given that our experimental approach was restricted to the use of CAF-derived CM. Interestingly, the resistance to PDT was only observed in the A431 cell line, but not in SCC13. Based on previous literature [22,34,36], we demonstrated that the TGFβ1 component of the CAF-derived CM was responsible for the induction of resistance to PDT in A431 cells, as this effect was abrogated by inhibiting TGFβ signalling through the use of the SB525334 TGFβ1 receptor I inhibitor, which has been extensively used in previous studies [41]. Complementary assays using recombinant TGFβ1 recapitulated the decrease in PDT toxicity in A431 cells and confirmed that this cytokine is capable of driving resistance to MAL-PDT in cSCC cells.

Our results have shown that the TGFβ1 secreted by CAFs is capable of inducing resistance to PDT in both bidimensional and three-dimensional A431 cell cultures. In this sense, migration assays using A431 spheroids showed that CAF-derived CM could also promote EMT, since increased explant expansion was observed from A431 spheroids treated with CAF-derived CM in combination with PDT compared to PDT alone. This results are in line with existing literature [44] and a deeper characterization of this aspect will need to be addressed in future studies. Overall, these data broaden the implications of our findings, as the use of multicellular spheroids is a powerful tool that more accurately mimics the in vivo events.

The fact that the effect of resistance to PDT was observed in A431 but not in SCC13 cells poses the question of intrinsic differences between the human-originated cSCC cell lines used in this study, even though both are widely used as in vitro models for the study of this type of cancer [37]. In fact, we found significantly different levels of endogenous TGFβ1 secreted by these two cSCC cell lines. In line with this, several authors have reported an association of the prodution of TGFβ1 with the response to PDT. More specifically, this cytokine induces an overexpression of HO-1 through the upregulation of P38 MAPK and PI3K pathways [45,46]. In agreement with this, our results indicate that higher secretion levels of endogenous TGFβ1 correlate with incresed expression of HO-1 in SCC13 cells.

In this sense, other authors have reported that dysregulations affecting the components of the heme synthesis pathway can compromise the optimal functioning of PDT [47], which are the ultimate effectors of PDT [13]. In this regard, the reduced basal ROS levels found in SCC13 cells may contribute to the intrinsic relative resistance to PDT displayed by this cell line and limit the susceptibility to develop increased resistance to PDT in response to exogenous TGFβ1. Accordingly, some authors have evidenced that higher secretion levels of endogenous TGFβ1 can reduce the sensitivity to exogenous TGFβ1, probably due to the saturation of the pathway by the autocrine signalling [44,48,49]. Hence, the high secretion levels of TGFβ1 found in SCC13 cells could further explain the lack of resistance to PDT observed in this cell line in the presence of TGFβ1. In contrast, the relatively higher basal ROS levels found in A431 cells are in line with the reduced endogenous TGFβ1 production and comparatively low expression of HO-1. Altogether, these results help to interpret the molecular mechanisms driving the higher sensitivity of A431 cells to PDT as well as the susceptibility of this cell line to develop resistance to PDT in the presence of exogenous TGFβ1.

The development of resistance to therapies in response to TGFβ1 has been tightly related by other authors to the slowing down of cell cycle progression [22,36,44,48,50]. In agreement with this, our data suggest that A431 cells undergo a decrease in proliferation upon the exposure to TGFβ1, evidenced by a shift into more flattened morphologies and the reduction in the mitotic index. Indeed, the cell cycle arrest has been reported as an effect of TGFβ1 and linked to resistance to therapies in previous studies [22,35,51]. Taken together, these data support the fact that TGFβ1 is responsible for the resistance to PDT caused by CAFs in A431 cells, likely acting through the induction of cell quiescence. On the other hand, neither changes in the mitotic index nor resistance to PDT were triggered in SCC13 cells by the CM from CAFs, and the threshold concentration of recombinant TGFβ1 needed to decrease cell proliferation in this cell line was shown to be higher than in A431 cells.

The fact that exogenous TGFβ1, even at relatively high concentrations, is able to reduce cell proliferation in SCC13 cells but does not lead to resistance to PDT may seem controversial. In this regard, our data point to the altered expression of components of the heme synthesis pathway, such as HO-1, as responsible for the intrinsic resistance to PDT of this cell line. This could explain why the expected effects of TGFβ1 in SCC13 are observed at the level of cell proliferation but not in the response to the PDT. Altogether, these results are consistent with a refractory phenotype of SCC13 cells that prevents the acquisition of resistance to PDT in response to TGFβ1. In addition, the sustained exposure to autocrine TGFβ1 signalling, as a result of the elevated endogenous production of this cytokine by SCC13 cells, may contribute to the reduced responsiveness to TGFβ1, in comparison to A431 cells. Nevertheless, there may be other important alterations involved. For this reason, future perspectives are focused on investigating the specific characteristics of different cSCC cell lines in relation to the TGFβ1 pathway and its mediators, as well as other players potentially involved in the regulation of the cell cycle progression and in the resistance to PDT.

## 5. Conclusions

In sum, we conclude that TGFβ1 secreted by CAFs can induce resistance to PDT through the induction of quiescence in cSCC cells. In addition, our results suggest that intrinsic features of different cSCC cell lines, including their own production of TGFβ1, can be determinant to predict the suitability of PDT as well as to define the window to modulate the effectiveness of this treatment by targeting CAF-derived TGFβ1. In sum, our work highlights CAF-derived TGFβ1 as a target for the optimization of PDT and paves the way to improved personalized treatments of particular cSCC tumors.

## Figures and Tables

**Figure 1 cancers-13-05613-f001:**
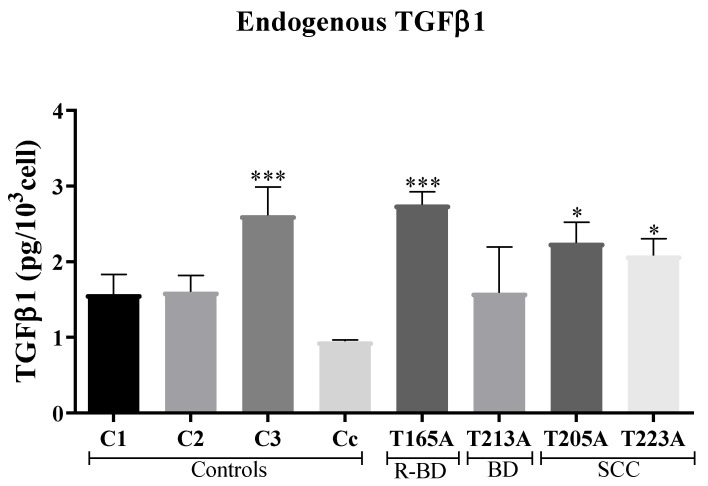
Secretion of endogenous TGFβ1 produced by fibroblasts. ELISA assays were performed to quantify the levels of TGFβ1 secreted to the culture media. In general, CAFs secreted higher levels of TGFβ1 than control fibroblasts. All the results are relativized to C1. Error bars correspond to S.E.M. (*n* = 3, one-way ANOVA, *: *p* < 0.05; ***: *p* < 0.001). R-BD: resistant Bowen disease; BD: Bowen disease; SCC: squamous cell carcinoma.

**Figure 2 cancers-13-05613-f002:**
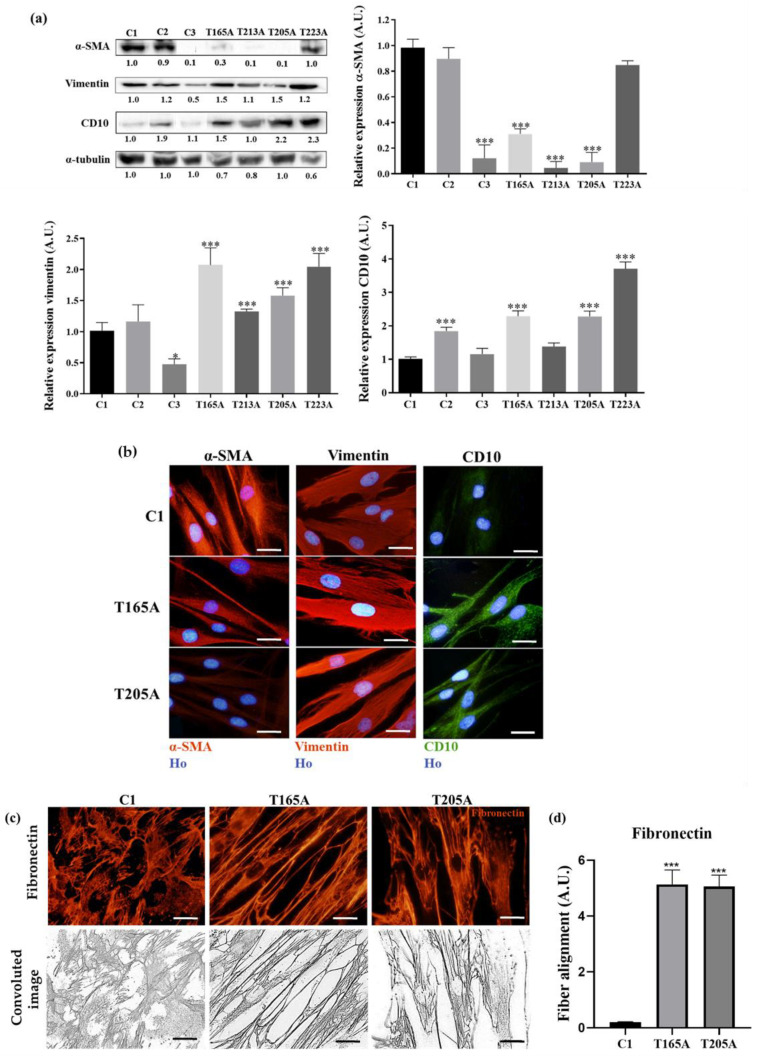
Expression of CAF markers in fibroblasts. (**a**) The expression levels of different markers (α-SMA, vimentin, and CD10) in fibroblasts was studied by WB, using α-tubulin as a loading control. All the results are relativized to C1. Error bars denote S.E.M. (*n* = 7, one-way ANOVA, *: *p* < 0.05; ***: *p* < 0.001). (**b**) The expression levels of α-SMA, vimentin, and CD10, and the distribution of fibronectin, were analysed by IF. Hoechst 33258 (Ho) as used for nuclear counterstaining. Scale bar: 20 μm. (**c**) The distribution of fibronectin was analyzed by IF (top row) and further image-processing using the ImageJ software (bottom row) (version 6.05, GraphPad Software Inc., San Diego, CA, USA) revealed an increase in the degree of alignment of the fibronectin fibers in CAFs (T165A and T205A) compared to control fibroblasts (C1). Scale bar: 20 μm. (**d**) Quantification and statistical comparison of the fibronectin fiber alignment in C1, T165A, and T205A. Error bars denote S.E.M. (*n* = 4, one-way ANOVA, ***: *p* < 0.001).

**Figure 3 cancers-13-05613-f003:**
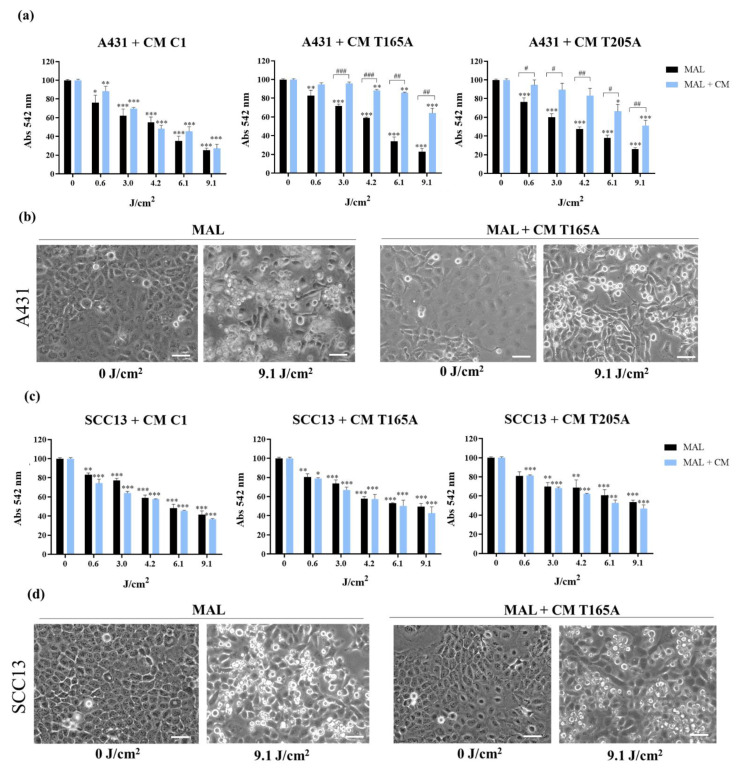
Effect of CAF-derived CM on the response of bidimensional cSCC cultures to PDT. (**a**) Cell viability rates of A431 cells treated with MAL-PDT (0.5 mM of MAL and red light doses between 0.6 and 9.1 J/cm^2^) in the presence or in the absence of CM from either control fibroblasts or CAFs. The results of the MTT assay indicate that CAF-derived CM (T165A, T205A) induced resistance to PDT, whereas this effect was not induced by CM from control fibroblasts (C1). (**b**) Phase contrast images illustrating the morphological change in morphology directly induced by the incubation of CAF-derived CM, showing an increased frequency of flattened cells in A431 cells incubated with CM from T165A (MAL + CM T165A, 0 J/cm^2^) compared with the control without CM (MAL, 0 J/cm^2^). After the treatment with PDT, lower rates of cell death were observed in the presence of CM (MAL + CM T165A, 9.1 J/cm^2^ compared with MAL + 9.1 J/cm^2^). Scale bar: 50 μm. (**c**) Cell viability of SCC13 cells in response to MAL-PDT in the presence or in the absence of CM from either control fibroblasts or CAFs, evaluated by MTT. (**d**) Phase contrast images of SCC13 cells exposed to MAL-PDT in the presence or in the absence of CM from CAFs, showing similar cell death rates. Control SCC13 cells reproduced the morphological changes observed in A431 cells in response to the treatment with CAF-derived CM alone. Error bars denote ± S.E.M. (*n* = 4, one-way ANOVA *: *p* < 0.05; **: *p* < 0.01; ***: *p* < 0.001; and *n* = 4, *t* test #: *p* < 0.05; ##: *p* < 0.01; ###: *p* < 0.001). Scale bar = 50 μm.

**Figure 4 cancers-13-05613-f004:**
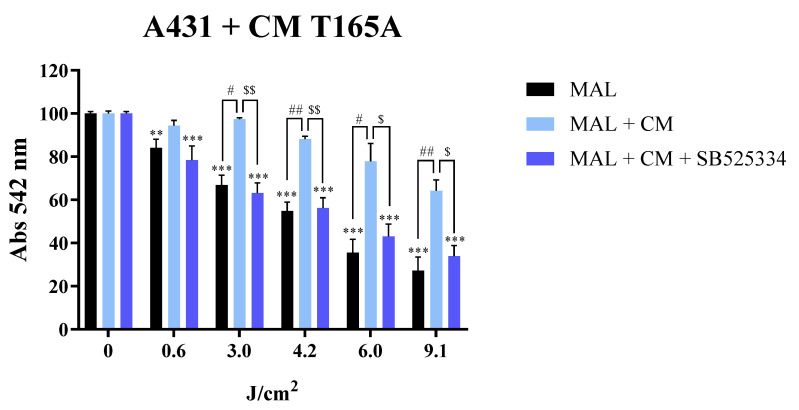
Abolishment of the resistance to PDT induced by CAF-derived CM in A431 cells by blocking TGFβ1 signalling. Cell viability of A431 cells pre-treated with 1 µM of SB525334 TGFβ1 receptor inhibitor and treated with MAL-PDT (0.5 mM of MAL and red light doses between 0.6 and 9.1 J/cm^2^) in the presence or in the absence of CM from CAFs. The results of the MTT assay indicate that the TGFβ1 present in CAF-derived CM (T165A) induced resistance to PDT. Error bars denote ± S.E.M. (*n* = 3, one-way ANOVA **: *p* < 0.01; ***: *p* < 0.001; *n* = 3, *t* test #: *p* < 0.05; ##: *p* < 0.01; and *t* test $: *p* < 0.05; $$: *p* < 0.01).

**Figure 5 cancers-13-05613-f005:**
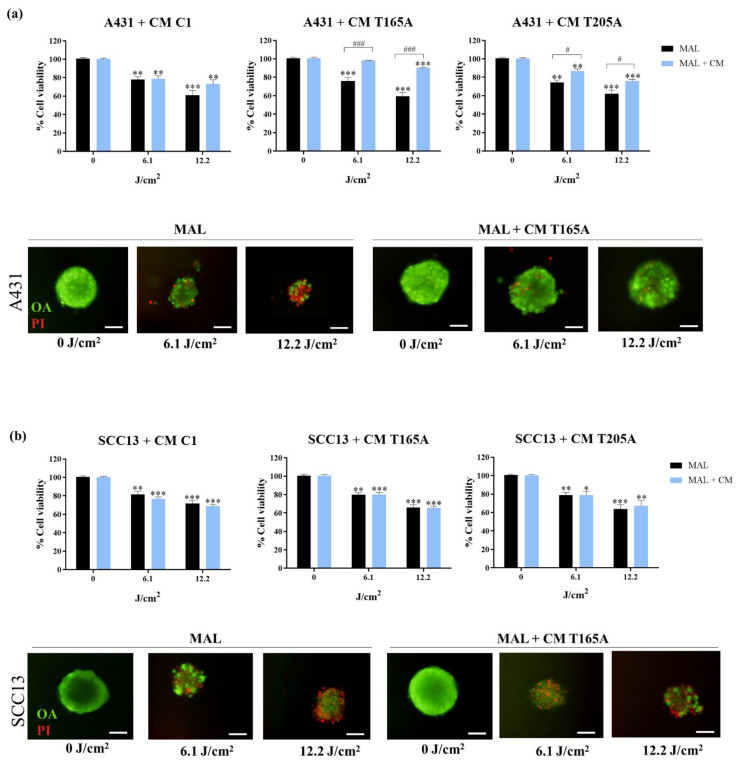
Effect of CAF-derived CM on the response of cSCC spheroids to PDT. (**a**) Cell viability in A431 spheroids in response to MAL-PDT (0.5 mM of MAL, red light doses from 6.1 to 12.2 J/cm^2^) in the presence of CM from control fibroblasts (C1) or CAFs (T165A, T205A), evaluated by a PI/OA assay. Resistance to PDT was induced by CM from CAFs but not from control fibroblasts. Top row: quatification. Bottom row: representative fluorescence microscopy images. (**b**) Cell survival in SCC13 spheroids in response to MAL-PDT in the presence of CM from CAFs or from control fibroblasts, evaluated by PI/OA assay. Top row: quantification. Bottom row: representative fluorescence microscopy images. Alive cells are visualized in green under blue light irradiation (450–490 nm); and dead cells in red under green light irradiation (570–590 nm). Error bars denote ± S.E.M. (*n* = 3, one-way ANOVA *: *p* < 0.05; **: *p* < 0.01; ***: *p* < 0.001; and *n* = 3, *t* test #: *p* < 0.05; ###: *p* < 0.001). Scale bar = 200 μm.

**Figure 6 cancers-13-05613-f006:**
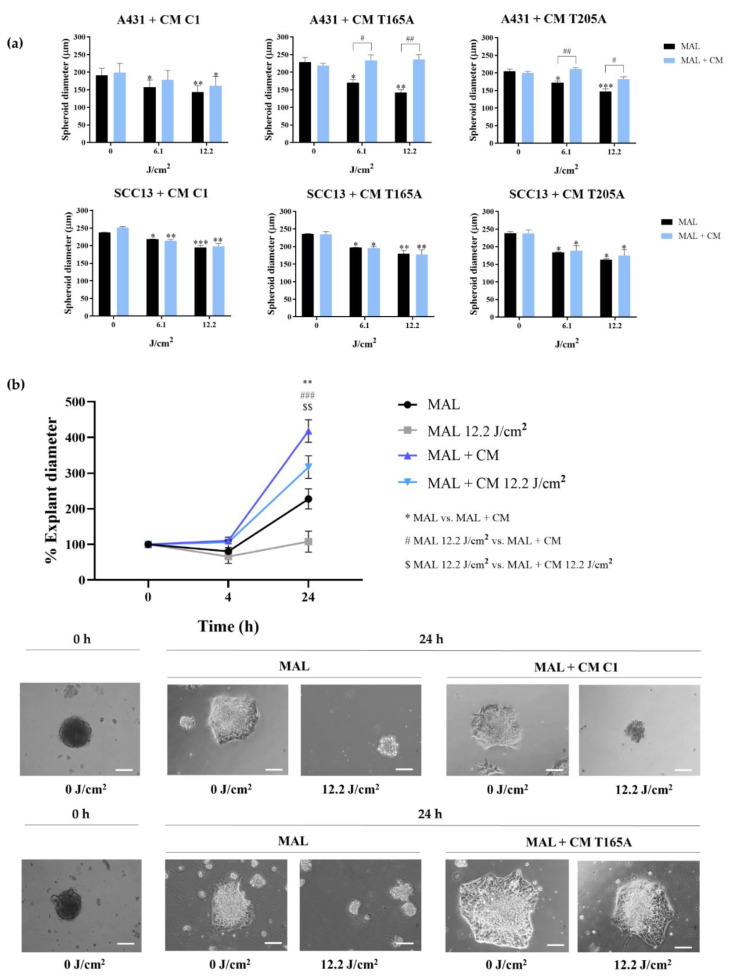
Effect of CAF-derived CM on the response of A431 spheroids to PDT. (**a**) Measurement of the diameter of A431 spheroids (top row) and SCC13 spheroids (bottom row) 24 h after MAL-PDT in the presence of CM from control fibroblasts or from CAFs. A431 spheroids in the presence of CAF-derived CM in combination with PDT were larger than those exposed to CM from C1 + PDT or to PDT alone, whereas SCC13 spheroids did not significantly change in size. (**b**) Measurement of the diameter of A431 explants in response to MAL-PDT (0.5 mM MAL, red light doses from 6.1 to 12.2 J/cm^2^) in the presence of CM from CAFs (T165A) or treated with PDT alone. A timecourse (0 h, 4 h, and 24 h) of the evolution of the explant diameter is shown. After 24 h, diameter of explants in presence of T165A CM was higher compared to PDT alone. (**b**) Representative phase contrast microscopy images of A431 explants, comparing control explants (MAL + 0 J/cm^2^) to those treated with different doses of PDT in the presence or in the absence of CAF-derived CM. Error bars denote ± S.E.M. (*n* = 3, one-way ANOVA MAL vs. MAL + CM *: *p* < 0.05; **: *p* < 0.01; ***: *p* < 0.001; *n* = 3, one-way ANOVA MAL 12.2 J/cm^2^ vs. MAL + CM #: *p* < 0.05; ##: *p* < 0.01; ###: *p* < 0.001; and one-way ANOVA MAL 12.2. J/cm^2^ vs. MAL + CM 12.2. J/cm^2^ $$: *p* < 0.01). Scale bar = 200 μm.

**Figure 7 cancers-13-05613-f007:**
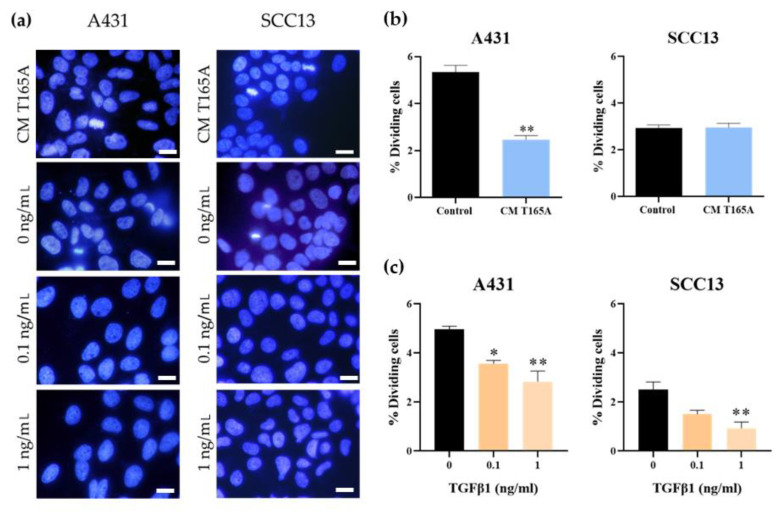
Effect of TGFβ1 on the mitotic rate of cSCC cell lines. The cells were incubated for 48 h with recombinant TGFβ1 or with CM from the T165A CAF culture before performing the analyses. (**a**) Representative images of cells stained with Hoechst 33258 (Ho) are shown. (**b**) The percentage of dividing cells was estimated after the treatment with recombinant TGFβ1. (**c**) The percentage of dividing cells was estimated after the treatment with CAF-derived CM. Error bars denote ± S.E.M. (*n* = 3, one-way ANOVA, *: *p* < 0.05; **: *p* < 0.01;). Scale bar = 20 μm.

**Figure 8 cancers-13-05613-f008:**
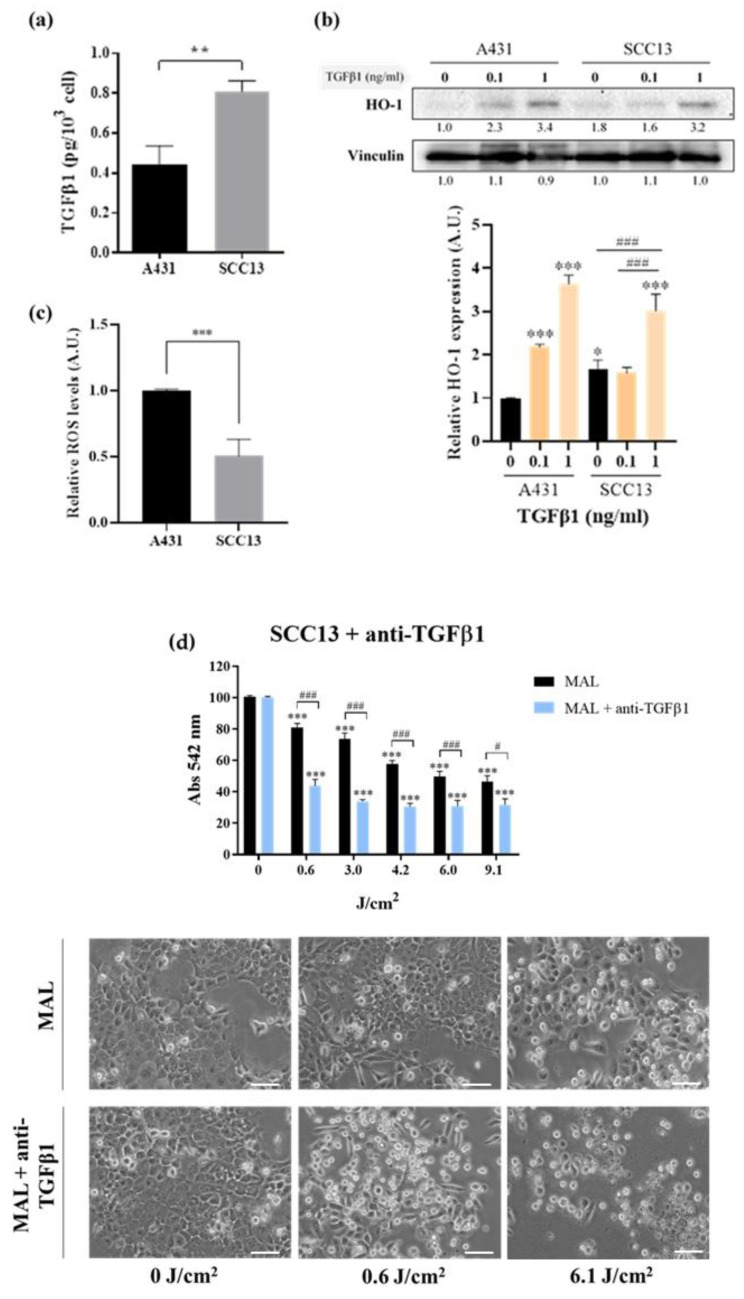
Molecular characterization of the cSCC cell lines. (**a**) Significant differences in the endogenous levels of secreted TGFβ1 were measured by ELISA, being higher in SCC13 compared to A431 cells. (**b**) The basal levels of ROS were evaluated by flow cytometry using DHF-DA, revealing significantly higher ROS levels in A431 compared to SCC13 cells. (**c**) The relative expression of HO-1 was evaluated by WB after the treatment with exogenous TGFβ1 at 0.1 and 1 ng/mL in both cell lines, using the corresponding untreated cells as a reference. The results indicate significantly higher basal expression levels of HO-1 in SCC13 compared to A431 cells. While both 0.1 and 1 ng/mL of exogenous TGFβ1 were sufficient to significantly increase the expression of HO-1 in A431 cells, a threshold of 1 ng/mL TGFβ1 was needed to induced increased HO-1 expression in SCC13. (**d**) Viability after MAL-PDT (0.5 mM of MAL and red light doses between 0.6 and 9.1 J/cm^2^) of SCC13 cells pre-treated with 3 µg/mL of a TGFβ1-neutralizing antibody. The MTT assay revealed that the intrinsic resistance to PDT of SCC13 cells was abolished by the inhibition of TGFβ signalling. Error bars denote ± S.E.M. (*n* = 3, one-way ANOVA, *: *p* < 0.05; **: *p* < 0.01; ***: *p* < 0.001; in (**b**), n = 3, one-way ANOVA comparison between 0, 0.1 and 1 ng/mL of exogenous TGFβ1, #: *p* < 0.05; ###: *p* < 0.001), in (**d**), *n* = 3, one-way ANOVA MAL vs. MAL + anti-TGFβ1, #: *p* < 0.05; ###: *p* < 0.001). Scale bar = 50 μm.

## Data Availability

Data may be obtained from the corresponding author on request.

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
