# Peer review of "TGFβ1 Secreted by Cancer-Associated Fibroblasts as an Inductor of Resistance to Photodynamic Therapy in Squamous Cell Carcinoma Cells"

_cancers, 2021, doi:10.3390/cancers13225613_

Round 1

Reviewer 1 Report

manuscript is corrected as recommended

now  it can be published

Reviewer 2 Report

no comment

Reviewer 3 Report

I would like to thank the authors for the extensive effort made to address my original review and I believe that with these new results the manuscript provides novel insight into how CAFs regulate the response to photodynamic therapy in cSCC.

This manuscript is a resubmission of an earlier submission. The following is a list of the peer review reports and author responses from that submission.

Round 1

Reviewer 1 Report

Manuscript needs major revision

Abstract should  be clear and concise

Abstracts  conclusion should  be specific  with main  finding

In the Introduction and the Discussion elaborate more  on Cutaneous squamous cell carcinoma  clinics, Surgery and chemotherapy, radio-therapy  and Photodynamic therapy

Table 1,2 are not  necessary,  exclude

Fig1   Include also  phosphorylated proteins alpha- SMA Vimetin and CD10 and  effect  of photodynamic therapy on these  phosphorylated proteins 

Fig 3   in each  group  show effect on alpha- SMA Vimetin and CD10 phosphorylated  proteins with total  proteins too

Fig 4  Legend include  detailed  explanation effect  to the Photodynamic therapy

Fig 5,6  are  not necessary  exclude

In  Discussion  correlate Photodynamic therapy to change in phosphorylated proteins and cancer cell  proliferation   and  migration

Reviewer 2 Report

This manuscript evaluated the influence of TGF-β1 secreted by cancer-associated fibroblasts (CAF) on the response to photodynamic therapy of SCC cell cultures. The authors reported evidence of resistance to PDT after TGF-β1 administration in one of the two SCC cell lines.

It is already known that TGF-β1 promotes heterogeneity and drug resistance in squamous cell carcinoma. However, the authors want demonstrate the implication of CAF as producers of TGF-β1 and consequently inducers of resistance to PDT-therapy. This result would be important for a new therapeutic strategy to optimize treatment with PDT of SCC, but the authors' conclusions are not fully supported by the results.

The first experimental part on CAF has been particularly taken care of, while the part on the SCC lines presents some deficiencies. The methods used to support their hypothesis and conclusion are limited. The authors should improve the experiments including functional and molecular assays with other cell lines.

In fact, there are contradictions in the opposite behavior of the two SCC cell lines  that need to be clarified with further experiments and with the confirmation of authors’ hypothesis  with the use of other SCC cell lines.

There are also contradictions in the methods used. For example, MTT assay is sometimes used to assess cell viability and other times to assess metabolism without implications on cell viability. This is confusing.

The treatments of the spheroids have been evaluated with few parameters: only the viability has been evaluated. No important parameters have been studied to assess the effects on spheroids, such as the area of the spheroid or migration or progression markers.

Reviewer 3 Report

This manuscript from Gallego-Rantero et al. describes a role for CAF-derived TGF-beta in the resistance of cSCC cells to photodynamic therapy. In general, this study is well presented and clearly described. However, I have some reservations about whether the conclusions drawn are sufficiently supported by the data presented, particularly given the variable phenotypes observed between fibroblast cultures from different patients and cell lines. I have described specific points that should be addressed to allay these concerns below.

  1. A key underlying principle of this study is that CAFs up-regulate TGF-beta. However, this is not convincingly demonstrated. Figure 2 shows that the 3/4 CAFs have increased endogenous TGF-beta compared to child foreskin controls (C1 and C2). I do not think that these cells are appropriate controls in this experiment, particularly given that the only adult fibroblast control analysed (C3) also has significantly increased endogenous levels of TGF-beta1 compared to C1 and C2.
    • The authors must demonstrate that endogenous TGF-beta1 levels are increased in CAFs compared to adult skin fibroblasts and provide an explanation for why this up-regulation was not found compared to C3. Ideally these experiments would be done using patient matched control tissue. If this is not possible adult human dermal fibroblasts are commercially available
  2. The role of TGF-beta in CAF-dependent cSCC cell resistance to PDT also requires further evidence. The authors show that CAF-derived conditioned media increases viability of A431 cells grown in 2D monolayers and as spheroids and suggest that this is due to TGF-beta within the conditioned media. However, this conditioned media will contain many different cytokines or growth factors that could be responsible for the observed increase in cell viability.
    • The authors should repeat these experiments with a TGF-beta blocking antibody or TGF-beta receptor inhibitor and show that this prevents the conditioned media from increasing cell viability.
  3. The impact of CAF-CM on cSCC viability is shown to be limited to A431 (and not in SCC13). The authors suggest that this is due to increased levels of endogenous TGF-beta in the SCC13 cells (I was not able to assess the validity of this this as the supplementary figure was not provided for review).
    • If this is the case, then the experiment suggested above should also show a further decrease in viability when TGF-beta is inhibited.
    • Furthermore, this conclusion is at odds with the data showing that SCC13 cells still show reduced metabolic activity and proliferation in response to exogenous TGF-beta. The authors must address this discrepancy. 
  4. The manuscript describes the elucidated mechanism as a general feature of cSCC. However, the data presented clearly shows that the phenotypes observed are highly variable. The data presented shows that TGF-beta up-regulation is not consistently observed in all CAFs and increased resistance to PDT is only observed in 1/2 cell lines tested. This has important consequences for how generalisable and clinically relevant the mechaism described is, impacting the interpretation of these results
    • The authors must clearly acknowledge throughout the manuscript that the effect they are describing is likely to be limited to a subset of cSCC tumours or pre-malignant lesions. They should also add details of how patients where this mechanism is active could be identified to the discussion section.